# Discrimination of Adzuki Bean (*Vigna angularis*) Geographical Origin by Targeted and Non-Targeted Metabolite Profiling with Gas Chromatography Time-of-Flight Mass Spectrometry

**DOI:** 10.3390/metabo10030112

**Published:** 2020-03-17

**Authors:** Tae Jin Kim, Jeong Gon Park, Soon Kil Ahn, Kil Won Kim, Jaehyuk Choi, Hyun Young Kim, Sun-Hwa Ha, Woo Duck Seo, Jae Kwang Kim

**Affiliations:** 1Division of Life Sciences, College of Life Sciences and Bioengineering, Incheon National University, Incheon 22012, Korea; f91gd@inu.ac.kr (T.J.K.); ToLuck@inu.ac.kr (J.G.P.); skahn@inu.ac.kr (S.K.A.); kilwon@inu.ac.kr (K.W.K.); jaehyukc@inu.ac.kr (J.C.); 2Division of Crop Foundation, National Institute of Crop Science, Rural Development Administration, Wanju, Jeonbuk 55365, Korea; hykim84@korea.kr; 3Department of Genetic Engineering and Graduate School of Biotechnology, Kyung Hee University, Yongin 17104, Korea; sunhwa@khu.ac.kr

**Keywords:** adzuki bean, geographical origin, metabolomics, multivariate analysis, non-targeted metabolite profiling

## Abstract

As international food trade increases, consumers are becoming increasingly interested in food safety and authenticity, which are linked to geographical origin. Adzuki beans (*Vigna angularis*) are cultivated worldwide, but there are no tools for accurately discriminating their geographical origin. Thus, our study aims to develop a method for discriminating the geographical origin of adzuki beans through targeted and non-targeted metabolite profiling with gas chromatography time-of-flight mass spectrometry combined with multivariate analysis. Orthogonal partial least squares discriminant analysis showed clear discrimination between adzuki beans cultivated in Korea and China. Non-targeted metabolite profiling showed better separation than targeted profiling. Furthermore, citric acid and malic acid were the most notable metabolites for discriminating adzuki beans cultivated in Korea and China. The geographical discrimination method combining non-targeted metabolite profiling and pareto-scaling showed excellent predictability (*Q*^2^ = 0.812). Therefore, it is a suitable prediction tool for the discrimination of geographical origin and is expected to be applicable to the geographical authentication of adzuki beans.

## 1. Introduction

The adzuki bean (*Vigna angularis*), which contains starch, nutritious proteins, and vitamins and has a sweet taste, has been used as an ingredient in traditional dessert foods, particularly in East Asian countries such as Korea, China, and Japan [1,2,3]. Adzuki beans have been extensively cultivated in more than 30 countries worldwide [4,5,6]. Additionally, international food trade has dramatically increased owing to the development of food storage and transportation technologies. Therefore, consumers are exposed to foods with foreign geographical origins through imports and have consequently become interested in the safety and authenticity of their food, which are linked to geographical origin [7]. Thus, European countries have placed country-of-origin labels on agricultural foods since September 2000 [8]. The adzuki bean is cooked together with glutinous rice to produce red rice dishes in Korea. Due to the mass cultivation of adzuki bean in China, large quantities of adzuki bean have been imported from China to Korea [9,10]. More than 80% of adzuki beans consumed in Korea are imported from China. Identification of the geographical origin of adzuki bean is critical because some local vendors deceive customers regarding cultivation origin for economic reasons. Tools for the validation of food safety and authenticity have become essential and have been developed using various analytical and genomic methods for the discrimination of geographical origin [1,7,11,12,13]. Although the genomic method is highly accurate, it cannot discriminate against the geographical origins of the same plant variety [14]. On the other hand, analytical methods such as liquid chromatography (LC)–mass spectrometry (MS) and gas chromatography (GC)–MS allow for accurate determination of the geographical origins of the same variety by analysing differences in chemical composition. However, there are currently no analytical methods based on metabolomics for the precise and accurate discrimination of the geographical origin of adzuki beans.

Metabolomics has been performed to differentiate the geographical origins of many foods, such as green tea, grape berry, wine, *Angelica gigas*, tobacco, cabbage, olive oil, wheat, pork, tomato, and coffee [15,16,17,18,19,20,21,22,23,24]. Several techniques, including GC–MS, LC–MS, capillary electrophoresis (CE)–MS, and nuclear magnetic resonance (NMR) spectroscopy, allow for accurate determination of the geographical origins of the same variety by analysing differences in chemical composition. Among these techniques, GC–MS represents a relatively robust and inexpensive analytical system. Previously, we used GC–time-of-flight (TOF) MS to investigate the possibility for metabolic discrimination between *Tagetes* cultivars [25]. Putri et al. [21] reported an application of non-targeted GC–MS metabolomics for the discrimination analysis of the geographical origin of coffee samples. Recent analytical techniques for non-targeted metabolite profiling are essential for generating large-scale data to obtain a comprehensive understanding of samples. Targeted metabolite profiling is used to identify and quantitate metabolites, which have already been confirmed using standard compounds, for example, to provide information towards understanding specific metabolic pathways and quality assessment [26,27]. Non-targeted metabolite profiling is used more globally, for example, to discover biomarkers [28,29]. To perform targeted metabolite profiling, the retention time and mass spectral information of the metabolites from standard compounds are required for identification and quantitation. Non-targeted metabolite profiling uses global libraries (such as NIST, Wiley, and HMDB) to identify peaks but cannot provide accurate quantification. However, non-targeted metabolite profiling can distinguish both known and unknown metabolites and quickly and reliably identify the peaks. Thus, the non-targeted approach can provide more comprehensive metabolite profile data than targeted profiling, but the data from targeted metabolite profiling are more accurate, sensitive, and quantitative. Non-targeted metabolomics has been performed using LC–TOFMS and GC–TOFMS [28,30,31,32,33]. GC–TOFMS commonly provides profiles of primary metabolites with good reproducibility. An advantage of GC–TOFMS is the use of standardised metabolome libraries to identify metabolites [32,34,35]. GC–TOFMS results can be directly matched with spectral libraries, whereas LC–TOFMS requires further validation for definite metabolite identification. Previously, we determined primary metabolism interplay using GC–TOFMS-based metabolite profiling in oval- and rectangular-shaped Chinese cabbage [22]. Furthermore, Zhao et al. [20] successfully used GC–MS metabolomics for the investigation of geographical location-associated primary metabolic changes in tobacco plants.

Metabolomics, combined with multivariate statistical analysis, has been used for many analyses, such as the discrimination of metabolic phenotype and geographical origin and the analysis of relationships between samples and metabolites [27,30,32]. To extract notable information from large-scale metabolite profile datasets, appropriate multivariate statistical analysis is essential. Before performing multivariate statistical analysis of complex metabolite profile data, the data should be pre-treated according to the intended purpose [32,36]. For example, principal component analysis (PCA) focuses on explaining as much of the variation in the data as possible. The discrimination of phenotype or geographical origin usually uses PCA, although the analysis should focus on differences among samples. Therefore, with the use of unit variance (UV)-scaling, which sets the standard deviation of all variables to 1 and thus makes all variables equally important, the transformed data may enhance the PCA results, whereas discriminant analysis may make the contribution of variables to the discrimination unclear. Thus, the choice of pre-treatment methods, such as UV-scaling, pareto-scaling, and range-scaling, is important to enhance the results of multivariate statistical analysis.

The objective of the present study is to discriminate the geographical origin of adzuki beans. For this purpose, multivariate statistical analysis (PCA, orthogonal partial least squares discriminant analysis (OPLS–DA)) was performed with targeted and non-targeted metabolite profiling using GC–TOFMS. To enhance the results of multivariate statistical analysis, UV-scaling and pareto-scaling were compared as pre-treatment methods. The final goal is the development of the first reliable geographical origin discrimination method for adzuki beans.

## 2. Results 

### 2.1. Comparison of Targeted and Non-Targeted Metabolite Profiling Using GC–TOFMS

To discriminate the geographical origin of adzuki beans, we analysed hydrophilic adzuki bean components using targeted and non-targeted metabolite profiling. First, targeted metabolite profiling was performed with GC–TOFMS. We detected 36 hydrophilic compounds in 13 different adzuki bean cultivars. The compounds were confirmed using standards and the in-house libraries NIST 11 and Wiley 9. Furthermore, the same data files obtained from targeted metabolite profiling were analysed using non-targeted metabolite profiling, and we identified 111 compounds. For the non-targeted approach, we used the statistical compare package of ChromaTOF software (LECO, St. Joseph, USA), and the data processing cut-offs were S/N (signal-to-noise ratio) ≧10 and mass spectral minimum similarity match ≧700.

The retention times obtained from non-targeted metabolite profiling were compared with those from targeted profiling to assess the accuracy of peak identification (Figure 1). The targeted and non-targeted metabolite profiling approaches identified 19 compounds in common with the same retention times. Despite the automatic identification of results using the statistical compare package for the non-targeted metabolite profiling platform, correlation of the retention times of both platforms showed a correlation coefficient of 1.000 (Figure 2). Furthermore, comparing adzuki bean chromatograms between targeted and non-targeted platforms showed the same retention times. The integration results of both platforms were also calculated to be almost the same (Appendix A). 

### 2.2. Geographical Discrimination of Adzuki Beans Using Multivariate Statistical Analysis with UV-Scaling

To discriminate the geographical origin of adzuki beans, the obtained data matrices from targeted and non-targeted metabolite profiling were subjected to multivariate statistical analysis (PCA and OPLS–DA), which was used to identify the features of samples occurring in each data matrix. PCA, an important tool for identifying overall patterns in complex data matrices, uses an orthogonal linear transformation to convert the original data into a new set of variables called principal components (PCs). The PC scores and loading are represented as a bi-dimensional plot and can indicate patterns in a dataset generated from samples. The data were normalised with UV-scaling. The PCA results of both platforms showed no variances among the samples (Appendix A).

To optimise the separation among samples, we used OPLS–DA to determine differences in metabolites arising from the geographical origin. OPLS–DA is an extension of the supervised partial least squares regression method in which features (X variables) are divided to separate the systematic variation into two parts, one that models the correlation between X and Y (prediction) and another that models the orthogonal (uncorrelated to Y) components [37]. Thus, OPLS–DA has maximum separation by classes of observations based on their variables and shows better interpretability compared with PLS–DA. The geographical origins were set to 1 for Korea and 2 for China. To validate the model, an internal validation method was used. The *R*^2^ and *Q*^2^ values of the validation results indicate the quality of the model. *R*^2^, the goodness of fit, indicates what proportion of variation in the data is explained by the model, and *Q*^2^, the goodness of prediction, indicates what proportion of variation in the data is predictable by the model. Both *R*^2^ and *Q*^2^ have a minimum of zero and a maximum of one. An *R*^2^ value closer to 1 is desirable; *Q*^2^ > 0.5 indicates a good prediction model, and *Q*^2^ > 0.9 indicates an excellent prediction model. The OPLS–DA projection models of both platforms showed good separation. The prediction model from targeted metabolite profiling showed *R*^2^X of 0.359, *R*^2^Y of 0.774, and *Q*^2^ of 0.638. The *Q*^2^ above 0.50 indicates a good prediction model (Table 1). In addition, the prediction model of non-targeted metabolite profiling platform showed *R*^2^X of 0.219, *R*^2^Y of 0.900, and *Q*^2^ of 0.777; this model also had a good prediction ability. The score plots of the targeted and non-targeted metabolite profiling data showed separation by geographical origin (Korea and China) (Figure 3A,B). In addition, the *R*^2^Y and *Q*^2^ values of the non-targeted metabolite profiling platform were higher than those of the targeted platform, but the *R*^2^X value was lower (Table 1). These results mean that the classes (Y) had more influence on explanation and prediction in the model than the variables (X) in the OPLS–DA results of the non-targeted metabolite profiling platform. 

Based on the above-discussed results, the non-targeted metabolite profiling score plot showed a clearer separation by geographical origin than targeted metabolite profiling (Figure 3A,B). The loading plots of both platforms had similar separation patterns. In the loading plots of both platforms, OPLS 1 and OPLS 2 resolved the separation of adzuki bean geographical origin. The significant metabolites of OPLS 1 in the targeted loading plot were 26 (citric acid), 23 (phenylalanine), 19 (aspartic acid), and 25 (shikimic acid), for which the eigenvector values were −0.32861, −0.30755, −0.21853, and 0.25839, respectively (Figure 3A and Appendix A). The important metabolites of the non-targeted OPLS 1 loading plot were A20 (analyte 20), 26 (citric acid), 23 (phenylalanine), A4 (analyte 4), A28 (analyte 28), and 49 (d-(-)-erythrose), for which the eigenvector values were −0.22680, −0.19944, −0.16582, 0.16987, 0.19345, and 0.19569, respectively (Figure 3B and Appendix A). Especially, 23 (phenylalanine) and 26 (citric acid) were the most important metabolites in OPLS 1 from both platforms and contributed the most to the separation of Korean adzuki beans from Chinese adzuki beans. Furthermore, these metabolites were also top-ranked in variable importance in projection (VIP) plots from both platforms (Appendix A). The VIP values can be used to explain the contribution of metabolites to the projection, where VIP values greater than 1 indicate the greatest influence on the model. As a result, targeted and non-targeted metabolite profiling showed the same results from multivariate statistical analysis, thereby validating the accuracy of non-targeted metabolite profiling as equal to that of targeted profiling. In addition, non-targeted profiling had better *R*^2^ and *Q*^2^ values than targeted profiling, making it more suitable for the discrimination of geographical origin.

### 2.3. Geographical Discrimination of Adzuki Beans Using Multivariate Statistical Analysis with Pareto-Scaling

The above-discussed PCA and OPLS–DA results were obtained with data normalised based on UV-scaling. In UV-scaling, which is widely used for normalisation in machine learning algorithms and can handle various types of data [36,37], the mean is subtracted from each feature, and each feature is divided by its standard deviation. As a result, UV-scaling sets the standard deviation of all variables to 1 and all variables become equally important; however, this can emphasise noise in the data. Therefore, UV-scaling is useful for comparing metabolites based on correlations. However, for discriminating geographical origin, it is even more important to identify distinct metabolite differences between origins, for which pareto-scaling could be more suitable than UV-scaling. The pareto-scaling method is similar to UV-scaling but differs in that the scaling factor is the square root of the standard deviation. Therefore, all variables remain closer to the original data than in UV-scaling (Figure 4). Figure 4 shows the normalisation results of the non-targeted metabolite profiling data. The UV-normalised data did not retain the original data structure, whereas pareto-scaling retained the structure (Figure 4). Thus, prominent variables in the original data also have an impact on pareto-scaling normalised data. 

To compare UV-scaling and pareto-scaling, PCA and OPLS–DA were performed on the targeted and non-targeted metabolite profiling data using pareto-scaling normalisation (Appendix A and Figure 3C,D). The PCA score plot from pareto-scaling normalisation indicated no difference according to geographical origin in both platforms. These results were the same as the PCA results for UV-scaling.

The OPLS–DA score plot from pareto-scaling normalisation showed separation by geographical origin for both platforms (Figure 3C,D). The prediction model from the targeted metabolite profiling data using pareto-scaling normalisation had *R*^2^X of 0.595, *R*^2^Y of 0.668, and *Q*^2^ of 0.579 (Table 1). Using the non-targeted profiling data, the prediction model had *R*^2^X of 0.374, *R*^2^Y of 0.869, and *Q*^2^ of 0.812. Based on the *Q*^2^ values, both models had good predictive capabilities. The score plots of the OPLS-DA results from the targeted and non-targeted metabolite profiling data with pareto-scaling showed separation by geographical origin between Korea and China (Figure 3C,D). OPLS–DA of the non-targeted platform had higher *R*^2^Y and *Q*^2^ values than the targeted platform and therefore showed a clearer separation (Table 1). This result was the same as the OPLS–DA results of UV-scaling (Figure 3A,B). The loading plots of both platforms with pareto-scaling showed almost the same patterns (Figure 3C,D and Appendix A). OPLS 1 and OPLS 2 resolved the separation of adzuki bean geographical origin. The significant metabolites of OPLS 1 in the targeted loading plot were 26 (citric acid), 18 (malic acid), and 36 (raffinose), for which the eigenvector values were −0.89757, −0.24132, and 0.17323, respectively; the significant metabolites of OPLS 2 were 26 (citric acid), 18 (malic acid), 9 (phosphoric acid), and 35 (sucrose), for which the eigenvector values were −0.50135, 0.16589, 0.48967, and 0.66463, respectively (Figure 3C and Appendix A). The important metabolites of OPLS 1 in the non-targeted loading plot were 26 (citric acid), 18 (malic acid), 73 (D-galactose1), 8 (ethanolamine), and A28 (analyte 28), for which the eigenvector values were −0.77563, −0.25134, −0.20828, 0.10923, and 0.10447, respectively (Figure 3D and Appendix A); the significant metabolites for OPLS 2 were 26 (citric acid), 72 (d-glucopyranose), 18 (malic acid), and 35 (sucrose), for which the eigenvector values were −0.57343, −0.18603, 0.14331, and 0.61216, respectively. Notably, in each platform, 18 (malic acid) and 26 (citric acid) contributed the most to the separation of Korean adzuki beans from Chinese adzuki beans in both the OPLS 1 and 2 loading plots (Figure 3C,D; Appendix A). VIP plots of each platform showed that these metabolites had the greatest influence on the projection model (Appendix A). 

## 3. Discussion 

For the first time, the geographical origin of adzuki beans (Korean and Chinese) was discriminated using metabolomics and chemometrics. When comparing the OPLS-DA results of pareto-scaling with UV-scaling for targeted metabolite profiling, the *R*^2^X of pareto-scaling was higher, whereas *R*^2^Y and *Q*^2^ were slightly lower (Table 1). On the other hand, non-targeted metabolite profiling with pareto-scaling showed increased *R*^2^X and *Q*^2^ and slightly decreased *R*^2^Y compared with UV-scaling. 

The *R*^2^X values of pareto-scaling were higher than those of UV-scaling for both metabolite profiling platforms. An increase in *R*^2^X indicates that the explanatory power of the model by the variable (X) increases, and a decrease in *Q*^2^ indicates a decrease in the predictive power of the model. Pareto-scaling maintained the original data structure and spectral line shapes more effectively than UV-scaling, and thus the explanatory power of the model by variables (X) was higher. The *Q*^2^ values of pareto-scaling were higher than those of UV-scaling in the non-targeted metabolite profiling platform. On the other hand, the *Q*^2^ value of the targeted metabolite profiling platform was lower. The targeted metabolite profiling data matrix is smaller than that of non-targeted profiling to predict models by variables. Pareto-scaling of the targeted profiling data matrix resulted in a lower *Q*^2^ value than UV-scaling. In contrast, pareto-scaling of the non-targeted profiling data matrix had sufficient data, and therefore the *Q*^2^ value was higher than that from UV-scaling owing to the increase in *R*^2^X. In addition, PCA showed the same changes in *R*^2^X and *Q*^2^ between UV-scaling and pareto-scaling as those observed with OPLS–DA. 

In OPLS–DA, a decrease in *R*^2^Y indicates that the explanatory power of the model by the class (Y) decreases. This also occurs because a few important data points significantly contributed to account for the prediction of the model. However, the *R*^2^Y value of pareto-scaling was slightly lower in both platforms than UV-scaling, and non-targeted metabolite profiling had higher *R*^2^Y values than targeted metabolite profiling. This is because the non-targeted metabolite profiling had a larger dataset to explain and predict the model by the class (Y) than the targeted platform. Recently, a non-targeted GC-MS metabolic profiling method was used to classify specialty coffee from different geographical origins, in which pareto-scaling gave the best sample clustering [21]. In agreement with the previous study, the use of pareto-scaling and non-targeted metabolite profiling afforded the most predictable method for geographical origin discrimination of adzuki beans.

Moreover, to determine the accuracy of the OPLS–DA model of the non-targeted metabolite profiling with pareto-scaling, an external validation test was conducted (Figure 5). The 39 samples were divided into 27 training samples and 12 test samples. The Y class variables were set to 1 for Korea and 2 for China. The OPLS predictive models were constructed using the 27 training samples, and then the 12 test samples were projected on the established OPLS predictive model. The predicted results of the OPLS model from the external validation test displayed good discrimination of the geographical origin of adzuki beans with *R*^2^*X* = 0.371, *R*^2^*Y* = 0.868, and *Q*^2^ = 0.772. Furthermore, this OPLS model showed a root mean square error of prediction (RMSEP) of 0.226, which indicates accurate prediction since values closer to zero are desirable [38]. In addition, no adzuki bean samples cultivated in Korea and China were found to be on the borderline of 1.5, the threshold level, in the external validation test. Finally, in order to avoid the risk of over-fitting the OPLS model, a permutation test and analysis of variance of the cross-validated residuals (CV–ANOVA) were performed for this model. The permutation test was performed with 200 permuted models that were constructed with the use of randomised classification (Y) for the samples and provided a reference distribution of *Q*^2^ value for random data. This *Q*^2^ value was compared with the *Q*^2^ value of the original (unpermuted) OPLS model. If the *Q*^2^ value from the permutation test was smaller than the *Q*^2^ value of the real OPLS model, the model was regarded as a predictable model [39,40]. The result of the permutation test showed *Q*^2^ of −0.616; this was a lower value than the *Q*^2^ of the real OPLS model (Appendix A). The final validation test was performed with the use of a CV–ANOVA test to verify the validity of the model. The model was considered to be valid when the *p*-value was lower than 0.05 [37]. The *p*-value of the CV–ANOVA test was 0.0000011942.

A clear distinction in malic acid and citric acid concentration between Korea and China samples was observed. Several investigations have reported that malic acid and citric acid in plants varied in different cultivation regions [17,18,19,20,41]. The differences in environmental conditions might be the main factors in the different metabolite levels of different geographical origins. Pereira et al. [42] reported that malate content in grape berry was significantly related to light exposure. Malic enzyme activity increases between 10 and 46 °C. A study concerning the metabolic responses of tobacco plants to the environment has demonstrated that the intermediates of the TCA cycle display a clear negative correlation with environmental factors such as rainfall and temperature [20]. Therefore, this study suggests that malic acid and citric acid could be key metabolites for regional discrimination. In addition, we performed heat map visualisation of all the correlation coefficients with Pearson’s correlation analysis for metabolite–metabolite correlation associated with the discrimination of geographical origin. Figure 6 shows the two clusters, one that clustered organic acids, sugars, and shikimic acid pathway–related metabolites and another that grouped amino acids and sugar alcohols. The organic acids (such as citric acid, malic acid, oxalic acid, and succinic acid) and sugars (sucrose, glucose, fructose, and galactose) were important metabolites related to energy metabolism (glycolysis and the TCA cycle). Organic acids related to the TCA cycle are produced during photosynthesis, and they serve as carbon skeletons for amino acid biosynthesis and light-harvesting, respectively [18]. Thus, negative relationships were observed between carbon-rich primary metabolites and nitrogen-containing metabolites. 

In conclusion, the OPLS–DA results of targeted and non-targeted metabolite profiling platforms combined with UV-scaling and pareto-scaling showed clear separation of the geographical origin of adzuki beans from Korea and China. In addition, the results showed that 18 (malic acid) and 26 (citric acid) were the most notable metabolites for discriminating geographical origin and are thus potential candidates as biomarkers for Korean adzuki beans. Furthermore, when comparing data normalisation with UV-scaling and pareto-scaling, as well as targeted and non-targeted metabolite profiling platforms, *Q*^2^ was the highest with non-targeted profiling and pareto-scaling. Therefore, multivariate statistical analysis combined with non-targeted metabolite profiling and pareto-scaling is a suitable prediction tool for discriminating geographical origin. This discrimination platform consisting of non-targeted metabolite profiling combined with chemometrics based on pareto-scaling can potentially be applied to discriminating the geographical origin of other crops and foods. The metabolite contents of plants could be affected by genotype as well as environmental conditions. An interesting aspect of future research is to clarify the genotype × environmental interactions on the phytochemical composition in adzuki bean. Thus, future work involving a larger sample size from various production regions will be very important for geographical discrimination of adzuki beans.

## 4. Materials and Methods

### 4.1. Samples and Chemicals

Korean adzuki bean (*V. angularis*) sample cultivars (K1, Geomguseul; K2, Seona; K3, Yeonduchae; K4, Hongeon; K5, Hongjin; K6, Whinguseul; K7, Whinnarae) were grown at the National Institute of Crop Science, Rural Development Administration, Wanju-gun, Korea, during the 2018 growing season (May to November; Figure 7). Chinese adzuki bean samples (C1–C6) harvested from two regions (Harbin, Heilongjiang province and Yanji, Jilin province) in November 2017 were purchased from local markets in Xinzhou and Jiangxia (Wuhan city) provinces, China. In total, 13 adzuki bean samples were kept at −80 °C until required for analysis. Three biological replicates were prepared for each sample. Ribitol, *N*-methyl-*N*-trimethylsilyl trifluoroacetamide (MSTFA), and pyridine were purchased from Sigma-Aldrich (St. Louis, MO, USA). All other chemicals used in this study were reagent grade unless otherwise stated.

### 4.2. Extraction and Analysis of Hydrophilic Compounds

The extraction and analysis method used for hydrophilic compounds (amino acids, organic acids, sugars, and sugar alcohols) was described in a previous study [43,44]. One millilitre of a methanol:water:chloroform 2.5:1:1 (*v*/*v*/*v*) solution was added to a finely ground bean sample (0.01 g) for extraction. Ribitol (60 µL, 200 µg/mL) as an internal standard (IS) was added to the mixture and incubated using a Thermomixer Comfort (model 5355, Eppendorf AG, Hamburg, Germany) at 37 °C for 30 min at a mixing frequency of 1200 rpm. The mixed solution was centrifuged at 16,000× *g* for 3 min, after which 800 µL of the methanol/water phase was collected in a fresh tube and mixed with 400 µL of water. The methanol/water fraction was centrifuged at 16,000× *g* for 3 min, and 900 µL of the upper layer was pipetted into a fresh tube. The aliquots were evaporated for 2 h in a centrifugal concentrator (CC-105; TOMY, Tokyo, Japan), then freeze-dried for over 16 h. For derivatisation, 80 µL of 2% methoxyamine hydrochloride in pyridine (*w*/*v*) was added, and the mixture was incubated at 30 °C and 1200 rpm for 90 min using a thermomixer. Then, 80 µL of MSTFA was added, and the mixture was incubated at 37 °C and 1200 rpm for 30 min. GC–TOFMS analysis was performed using an Agilent 7890A gas chromatograph (Agilent, Atlanta, USA) coupled to a Pegasus HT TOF mass spectrometer (LECO) with a CP-SIL 8 CB column (30 m length, 0.25 mm diameter, and 0.25 µm thickness, Agilent). The split ratio, injector temperature, and helium gas flow were 1:25, 230 °C, and 1.0 mL/min, respectively. The column temperature was held for 2 min at 80 °C, increased at 15 °C/min to 320 °C, and then maintained for 10 min. The temperatures of the transfer line and ion source were 250 and 200 °C, respectively, and scan mode was used with a mass range of 85–600 *m*/*z*. Targeted metabolite profiling was performed as follows. In-house libraries were used to identify the metabolites using ChromaTOF software (LECO), the identities of which were also confirmed using standards (Appendix A). The Chroma TOF software package was used to extract raw peaks, filter and calibrate data baselines, align peaks, perform deconvolution analysis, identify peaks, and integrate peak areas. For relative quantification, we used ribitol as an IS, and ratios were calculated by the peak area of each metabolite based on that of the IS. As a result, a total of 36 metabolites were identified (i.e., Metabolomics Standards Initiative (MSI) level 1) and quantified [45,46,47] (see Appendix A). A similarity index of 70% or more was chosen as the cut-off value because it afforded 100% accuracy in analyte identification based on our previous experience (confirmed by co-injection of commercial standards) [45].

### 4.3. Non-Targeted Metabolite Profiling Data Processing

To compare targeted and non-targeted metabolite profiling, the same results obtained from targeted metabolite profiling were processed by non-targeted profiling. Chromatographic alignment and data processing for non-targeted metabolite profiling were performed using the statistical compare package of the ChromaTOF software (LECO). The following conditions were used: (1) the baseline was drawn just above the noise; (2) the signal-to-noise (S/N) cut-off for initial peak finding was set to 10 for a minimum of two apexing; (3) the maximum retention time difference was set to 2 s; (4) the mass spectral match score was ≥700; (5) unique masses were used for area and height calculations; (6) NIST 11, Wiley 9, and in-house libraries were used for searching. Relative quantification was calculated using the same method as that used for targeted metabolite profiling. As a result, a total of 111 metabolites were identified (i.e., MSI levels 1 to 3) and quantified (see Appendix A).

### 4.4. Statistical Analysis

All analyses were performed with three replicates. PCA and OPLS–DA (SIMCA-P version 13.0; Umetrics, Umeå, Sweden) were performed with data obtained from GC–TOFMS to discriminate the geographical origin of samples. To compare the effect of data normalisation in multivariate statistical analysis, all the data were normalised with UV-scaling and pareto-scaling. For UV-scaling, the mean is subtracted from each feature, and the values of each feature are divided by the standard deviation. Pareto-scaling is similar to UV-scaling, but the scaling factor is the square root of the standard deviation. PCA and OPLS–DA were based on the calculated eigenvectors and eigenvalues [34]. Score plots of the PCA and OPLS–DA results were used to discriminate the geographical origin of samples, and the loading and VIP plots demonstrated cluster separation of the samples in the score plots. Graphs of normalisation results using UV-scaling and pareto-scaling and correlation analysis were constructed using MetaboAnalyst 4.0 (https://www.metaboanalyst.ca). For the external validation test, the OPLS predictive models developed by OPLS–DA were used. The test set comprising approximately 30% (*n* = 12) of the entire samples was randomly chosen and was used for the external validation. The remaining samples were used for a training set (Korea: *n* = 15; China: *n* = 12). In the validation test, the response (Y) variable was set as 1 for Korean adzuki bean and 2 for Chinese adzuki bean, respectively. The limit value for the geographical classification of adzuki bean was set at 1.5, which is the average value between the response values of the respective groups (i.e., 1 and 2). The permutation test and CV–ANOVA were conducted by SIMCA-P version 13.0 (Umetrics).

## Figures and Tables

**Figure 1 metabolites-10-00112-f001:**
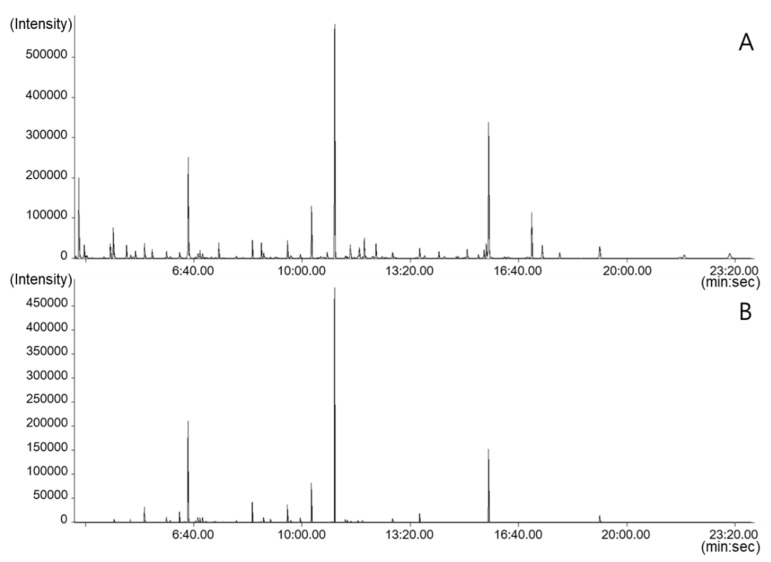
Analytical ion chromatograms of K5 adzuki bean samples obtained using non-targeted (**A**) and targeted (**B**) metabolite profiling with gas chromatography (GC)—time-of-flight (TOF) mass spectrometry (MS).

**Figure 2 metabolites-10-00112-f002:**
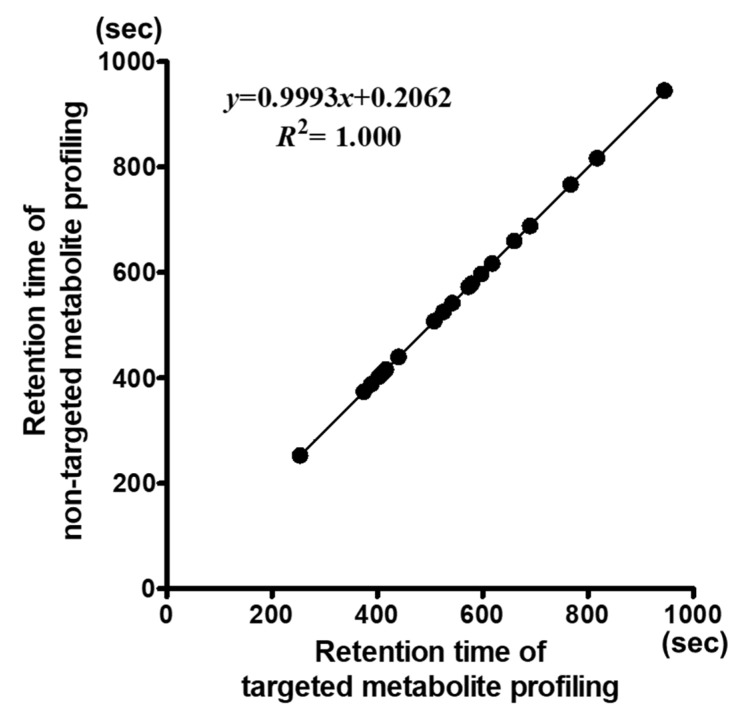
Retention time correlation of targeted and non-targeted metabolite profiling platforms. The *x*-axis shows the retention times (s) of 19 compounds detected by the targeted metabolite profiling method, and the *y*-axis shows those by the non-targeted metabolite profiling method.

**Figure 3 metabolites-10-00112-f003:**
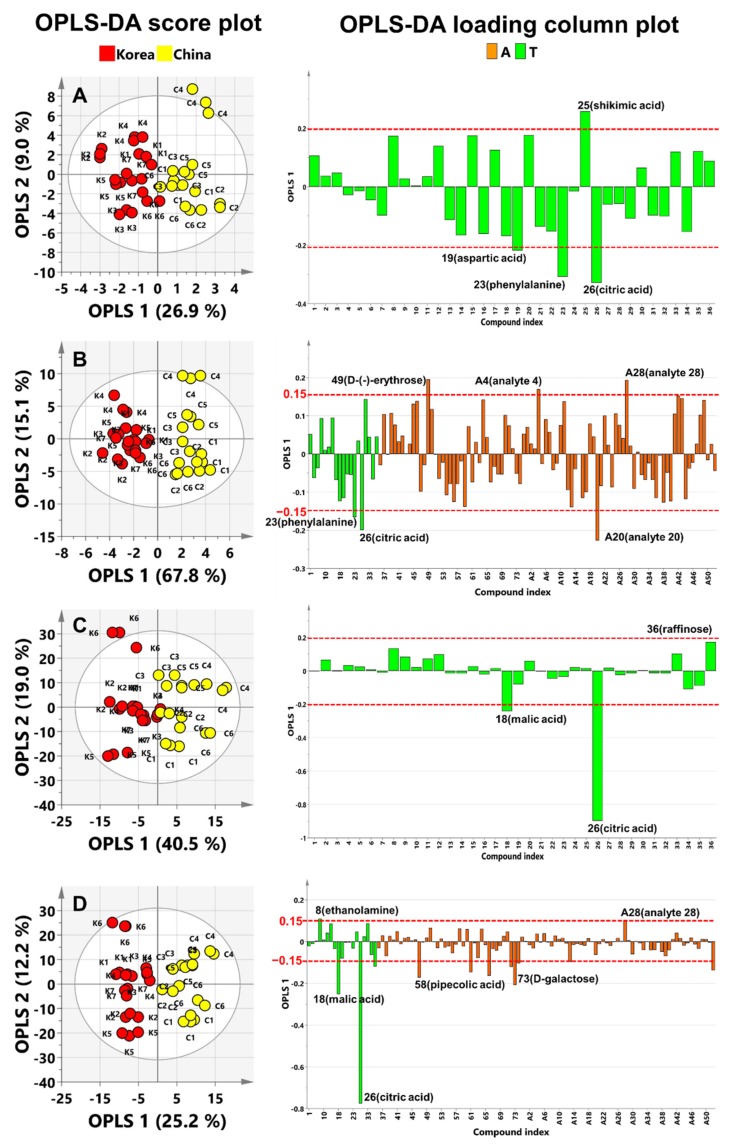
OPLS–DA score plots and loading column plots of targeted and non-targeted metabolite profiling platforms with UV-scaling and pareto-scaling normalisation. Score plots: (**A**), targeted, UV-scaling; (**B**), non-targeted, UV-scaling; (**C**), targeted, pareto-scaling; (**D**), non-targeted, pareto-scaling. Loading column plots: (**A**), automatically identified compounds by non-targeted metabolite profiling; T, identified compounds by targeted metabolite profiling.

**Figure 4 metabolites-10-00112-f004:**
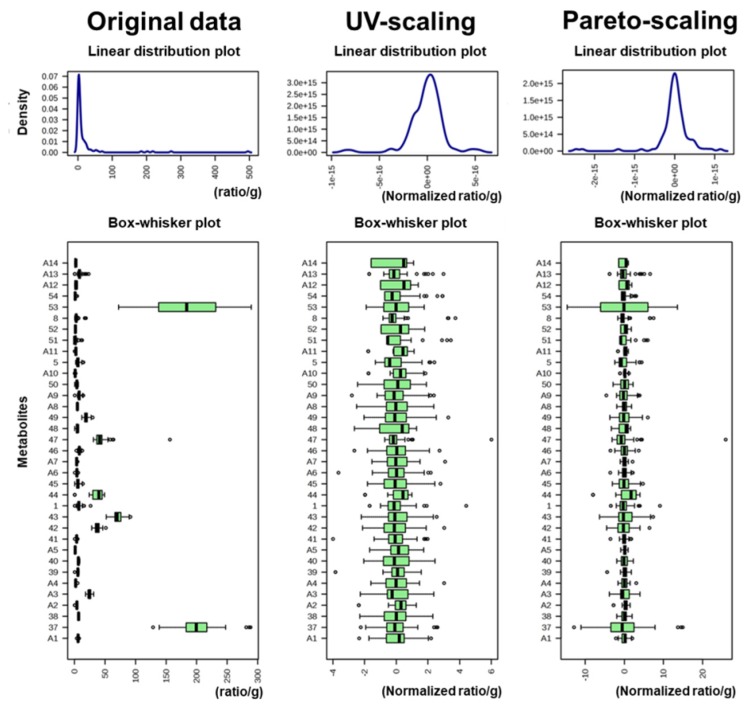
Normalisation results of non-targeted metabolite profiling data. Ratios were calculated by the peak area of each metabolite based on those of the internal standard (ribitol).

**Figure 5 metabolites-10-00112-f005:**
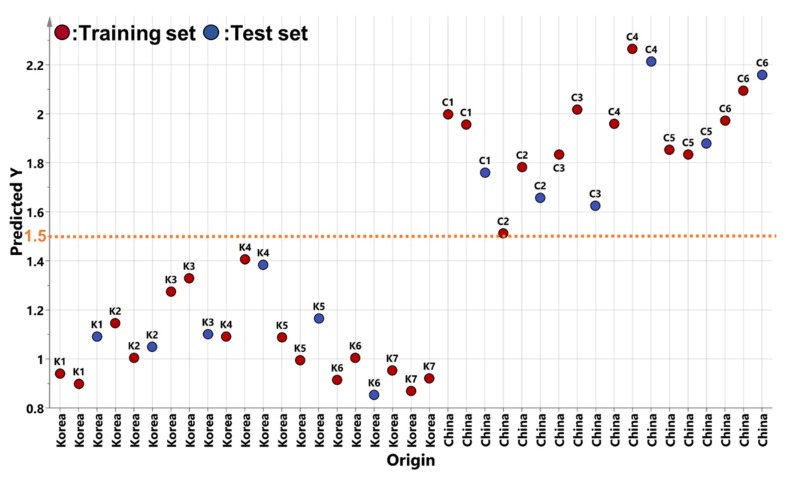
External validation test by OPLS–DA model of non-targeted metabolite profiling and pareto-scaling (*R*^2^*X* = 0.371, *R*^2^*Y* = 0.868, *Q*^2^ = 0.772, RMSEP = 0.226).

**Figure 6 metabolites-10-00112-f006:**
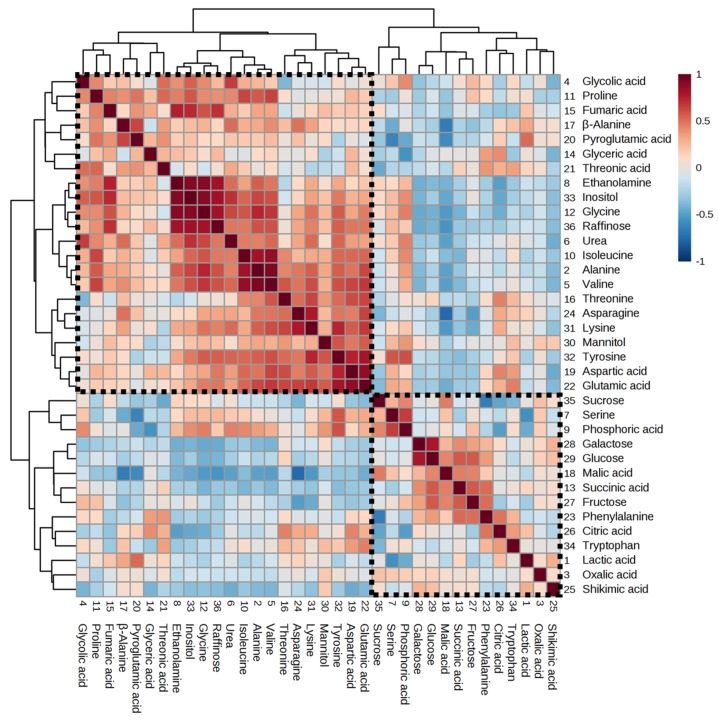
Correlation matrix of 36 metabolites in adzuki bean. Each square indicates a Pearson’s correlation coefficient for a pair of compounds. The value for the correlation coefficient is represented by the intensity of the blue or red colour, as indicated on the colour scale. Hierarchical clusters are presented as a cluster tree.

**Figure 7 metabolites-10-00112-f007:**
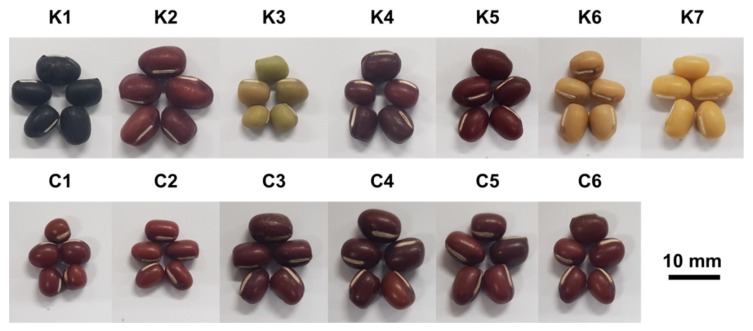
Variation in bean colour and shape among 13 adzuki bean phenotypes (K#, Korean sample number; C#, Chinese sample number).

**Table 1 metabolites-10-00112-t001:** Model validation results from multivariate statistical analysis (principal component analysis (PCA) and orthogonal partial least squares discriminant analysis (OPLS–DA)) of targeted and non-targeted metabolite profiling platforms with unit variance (UV)- and pareto-scaling.

Platform	Scaling	Model	*R*^2^X	*R*^2^Y	*Q* ^2^
Targeted	UV	PCA	0.421		0.182
	UV	OPLS–DA	0.359	0.774	0.638
	Pareto	PCA	0.634		0.133
	Pareto	OPLS–DA	0.595	0.668	0.579
Non-targeted	UV	PCA	0.328		0.130
	UV	OPLS–DA	0.219	0.900	0.777
	Pareto	PCA	0.491		0.167
	Pareto	OPLS–DA	0.374	0.869	0.812

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
