# Peer review of "Discrimination of Adzuki Bean (Vigna angularis) Geographical Origin by Targeted and Non-Targeted Metabolite Profiling with Gas Chromatography Time-of-Flight Mass Spectrometry"

_metabolites, 2020, doi:10.3390/metabo10030112_

Round 1
Reviewer 1 Report
This manuscript reports about the geographical difference of components measured by GC-TOF-MS in adzuki beans. The methodology of data normalization for multivariate statistical analysis in metabolome experiment is discussed. The authors concluded pareto-scaling was good for the statistical analysis method to discriminate components in adzuki bean. The methodology is clearly described and understandable. However, I have several question and suggestion comments as below.
The explanation about plant materials is obscure (L.277-285). Figure 6 represents appearance of all samples from Korea and China, totally 13 different cultivars were used. However, in Figure 3 and 5, the data plots are 21 for Korea and 18 for China respectively. The represented data includes sample/experimental replications. The same samples should be indicated in same symbols or marks. And, please add more information about adzuki bean samples and the experimental design in Materials and Methods section.
The sample production place in Korea is only one location (Wanju-gu, Korea). Furthermore, information of the production place of Chinese adzuki bean is missing. I feel it is impossible to discuss about the difference of geographical origin by these poor sampling locations and information.
I’m wondering that some change in components may be occurred in Chinese adzuki beans, because they were harvested in November 2017 and stored almost one year to be used for the experiment. I worry about the analytical results data may reflect the difference of freshness of beans. If adzuki beans are stored for long time in 4°C or room temperature, the color of seed coat is obviously changed (turned darker). That means the components of bean are significantly altered during the storage.
For the extraction of hydrophilic compounds form beans, authors used a method described in previous study of freeze-dried radish sprouts (Ref. #27). One milliliter of extraction solution and 0.01 g of finely ground bean sample were used. I feel the extraction volume scale is too small to detect the characteristics of geographical backgrounds. How many beans were ground to take 0.01 g sample? The 0.01 g is less than the weight of one adzuki bean.
Probably, because of the small scale of extraction volume, the componential differences between cultivars (varieties) are hardly detectable. Seven different colored Korean cultivars were used, however the difference among those seems to be too small. I think varietal differences should be more extent than geographical difference of production places.
In the discussion section, authors should mention about the reason why citric acid and malic acid were detected as notable metabolites to discriminate production places. What is the roles and functions of citric acid and malic acid in adzuki bean? How is content of these acids changed or physiologically regulated in the bean? Why are those contained amounts differed? A discussion paragraph about citric and malic acids with considerations of physiological aspects of adzuki bean is necessary.
None references are cited in the discussion section. The discussion should be made based on the related information from previous and other researches.
Author Response
This manuscript reports about the geographical difference of components measured by GC-TOF-MS in adzuki beans. The methodology of data normalization for multivariate statistical analysis in metabolome experiment is discussed. The authors concluded pareto-scaling was good for the statistical analysis method to discriminate components in adzuki bean. The methodology is clearly described and understandable. However, I have several question and suggestion comments as below.
[Comment] The explanation about plant materials is obscure (L.277-285). Figure 6 represents appearance of all samples from Korea and China, totally 13 different cultivars were used. However, in Figure 3 and 5, the data plots are 21 for Korea and 18 for China respectively. The represented data includes sample/experimental replications. The same samples should be indicated in same symbols or marks. And, please add more information about adzuki bean samples and the experimental design in Materials and Methods section.
[Response] We thank the reviewer for the comment. V. angularis samples were obtained from Wanju-gun (K1–K7) in Korea and Heilongjiang (C1–C2) province and Jilin province (C3–C6) in China. Thirteen adzuki bean samples collected from Korea and China were kept at -80 °C until required for analysis. Three biological replicates were prepared for each sample. This information has been added at the beginning of the Materials and Methods section in the revised manuscript. (Lines 333–340) Additionally, the labels K1–K7 and C1–C6 are shown in Figures 3 and 5.
[Comment] The sample production place in Korea is only one location (Wanju-gu, Korea). Furthermore, information of the production place of Chinese adzuki bean is missing. I feel it is impossible to discuss about the difference of geographical origin by these poor sampling locations and information.
[Response] As mentioned above, the production places were Harbin, Heilongjiang province (C1–C2) and Yanji, Jilin province (C3–C6), China (Line 337). However, we agree with the reviewer that these sampling locations are poor. We have added the following sentences in the revised manuscript to explain the reason for considering Korea and China in detail: “The adzuki bean is cooked together with glutinous rice to produce red rice dishes in Korea. Due to the mass cultivation of adzuki bean in China, large quantities of adzuki bean have been imported from China to Korea [9,10]. More than of 80 % of total adzuki bean consumed in Korea has been imported from China. Identification of the geographical origin of adzuki bean is critical because some local vendors deceive customers regarding cultivation origin for economic reasons.” (Lines 46–51) Thus, in this study, we used GC-TOFMS–based metabolite profiling combined with multivariate analysis to construct a discrimination model of geographical origin of adzuki beans samples collected from Korea (Wanju-gun) and China (Heilongjiang and Jilin provinces), for the first time. In addition, we used GC-TOFMS–based metabolite profiling combined with chemometrics to investigate biochemical reactions in adzuki bean samples because GC-TOFMS allows simultaneous detection of a diverse group of primary metabolites. We have added a biochemical discussion on the discriminant metabolites and Figure 6 in the revised manuscript (Lines 292–310). In this study, GC-TOFMS–based metabolomics could be used to discriminate geographical origin as well as to identify metabolic links in adzuki bean.
[Comment] I’m wondering that some change in components may be occurred in Chinese adzuki beans, because they were harvested in November 2017 and stored almost one year to be used for the experiment. I worry about the analytical results data may reflect the difference of freshness of beans. If adzuki beans are stored for long time in 4°C or room temperature, the color of seed coat is obviously changed (turned darker). That means the components of bean are significantly altered during the storage.
[Response] We have added the storage conditions to the revised manuscript. In total, 13 adzuki bean samples were kept at -80 °C until required for analysis. (Lines 338–340)
[Comment] For the extraction of hydrophilic compounds form beans, authors used a method described in previous study of freeze-dried radish sprouts (Ref. #27). One milliliter of extraction solution and 0.01 g of finely ground bean sample were used. I feel the extraction volume scale is too small to detect the characteristics of geographical backgrounds. How many beans were ground to take 0.01 g sample? The 0.01 g is less than the weight of one adzuki bean. Probably, because of the small scale of extraction volume, the componential differences between cultivars (varieties) are hardly detectable. Seven different colored Korean cultivars were used, however the difference among those seems to be too small. I think varietal differences should be more extent than geographical difference of production places.
[Response] We randomly selected 20 adzuki beans of each sample (approximately 3 g) and ground to a powder. GC-TOFMS is an extremely sensitive tool. We used 0.01 g of adzuki bean powder. In our previous studies, soybean residues were extracted with 1 ml of extraction solution and 0.01 g of finely ground sample (Foods 2020, 9, 117). In addition, Putri et al. (2019) performed a non-targeted GC-MS metabolic profiling to classify specialty coffee from different geographical origins and they used 0.015 g of coffee bean powder (Metabolomics, 15, 126).
However, we agree with the reviewer. We have included additional statements in the revised manuscript to highlight the importance of experimental design as follows: “The metabolite contents of plants could be affected by genotype as well as environmental conditions. An interesting aspect for future research is to clarify the genotype × environmental interactions on the phytochemical composition in adzuki bean. Thus, future work involving larger sample size from various production regions will be very important for geographical discrimination of adzuki bean.” (Lines 326–330)
[Comment] In the discussion section, authors should mention about the reason why citric acid and malic acid were detected as notable metabolites to discriminate production places. What is the roles and functions of citric acid and malic acid in adzuki bean? How is content of these acids changed or physiologically regulated in the bean? Why are those contained amounts differed? A discussion paragraph about citric and malic acids with considerations of physiological aspects of adzuki bean is necessary. None references are cited in the discussion section. The discussion should be made based on the related information from previous and other researches.
[Response] We have added the following sentences in the revised manuscript to discuss the reason for the discrimination markers (malic acid and citric acid) of adzuki bean and their origins in detail: “Several investigations have reported that malic acid and citric acid in plants varied in different cultivation regions (JAFC, 2009, 57, 1481–1490; JAFC, 2011, 59, 8806–8815; JAFC, 2013, 61, 2597–2605; JAFC, 2013, 61, 7994–8001). The differences in environmental conditions might be the main factors in the different metabolite levels of different geographical origins. Pereira et al. reported that malate content in grape berry was significantly related to light exposure (JAFC, 2006, 54, 6765–6775). Malic enzyme activity increases between 10 and 46 °C. A study concerning the metabolic responses of tobacco plants to the environment has demonstrated that the intermediates of the TCA cycle display clear negative correlation with environmental factors such as rainfall and temperature (Scientific Rep., 2015, 5, 16346). Therefore, this study suggests that malic acid and citric acid could be key metabolites for regional discrimination. In addition, we performed heat map visualization of all the correlation coefficients with Pearson’s correlation analysis for metabolite–metabolite correlation associated with the discrimination of geographical origin. Figure 6 shows the two clusters, one that clustered organic acids, sugars, and shikimic acid pathway–related metabolites and another that grouped amino acids and sugar alcohols. The organic acids (such as citric acid, malic acid, oxalic acid, and succinic acid) and sugars (sucrose, glucose, fructose, and galactose) were important metabolites related to energy metabolism (glycolysis and the TCA cycle). Organic acids related to the TCA cycle are produced during photosynthesis, and they serve as carbon skeletons for amino acid biosynthesis and light-harvesting, respectively (Scientific Rep., 2015, 5, 16346). Thus, negative relationships were observed between carbon-rich primary metabolites and nitrogen-containing metabolites.” (Lines 293–310)
Reviewer 2 Report
Although the results sound scientifically correct and conclusion are generally supported by the data, the manuscript has some flaws in the design that need amendment. My general comments are:
why only China and Korea have been considered, if several other asian countries produce adzuki beans? this should be eventually justified or integrated, to support the significance of the work the authors refer to targeted and untargeted (that are related to the MS acquisition) with the annotation level. I suggest to refer to COSMOS standards of reporting in metabolomics (or equivalent system) a large space has been left in both introduction and discussion relatively to normalization strategies; this part is rather known and should deserve less attention. ON the contrary, other aspects (such as metabolomics for food traceability, or the relevance of traceability for adzuki beans) would need more attention. Introduction should be re-written. why GC/TOF was chosen for this purpose? I expect that LC/QTOF targeting secondary metabolism would have been more effective in discrimination, because secondary metabolites are more affected by environment the choice of samples is questionable: korean samples are from the same site, i.e. they are not affected by pedo-climatic conditions, whereaas chinese samples are. On the contrary, cultivar is unknown for chinese samples but it is known for korean specimens. only 19 out of 36 compounds were confirmed by standard.. this sounds a little limiting, do the authors have an explanation? the ID process should be reported (monoisotopic mass? monoisotopic + isotope ratio?...) a biochemical discussion on the discriminant metabolites could be added hierarchical clustering might help dissecting cultivar effects from origin effects, once cultivar type is known why OPLS-DA was not the choice, in place of PLS-DA? in SIMCA you have this possibility.. this would help in limiting non predictive variability and offers the possibility of CV-ANOVA validation as well as permutation testing for overfitting line 217: silanol is likely an artefact of derivatization several errors in language are present throughout the whole manuscript, please revise for EnglishAuthor Response
Although the results sound scientifically correct and conclusion are generally supported by the data, the manuscript has some flaws in the design that need amendment. My general comments are:
[Comment] why only China and Korea have been considered, if several other asian countries produce adzuki beans? this should be eventually justified or integrated, to support the significance of the work
[Response] We thank the reviewer for the comment. To address this, we have included the following sentences in the revised manuscript: “The adzuki bean is cooked together with glutinous rice to produce red rice dishes in Korea. Due to the mass cultivation of adzuki bean in China, large quantities of adzuki bean have been imported from China to Korea [9,10]. More than of 80 % of total adzuki bean consumed in Korea has been imported from China. Identification of the geographical origin of adzuki bean is critical because some local vendors deceive customers regarding cultivation origin for economic reasons.” (Line 46–51)
[Comment] the authors refer to targeted and untargeted (that are related to the MS acquisition) with the annotation level. I suggest to refer to COSMOS standards of reporting in metabolomics (or equivalent system)
[Response] We described information of annotation level (Metabolomics Standards Initiative, MSI level) in supplementary tables. The targeted metabolites profiling is MSI level 1, while the non-targeted metabolites profiling is MSI level 1 to 3. We offered raw data matrix of targeted and non-targeted metabolite profiling from adzuki beans as supporting information for COSMOS standards reporting and cited papers related to COSMOS standards of reporting in the revised manuscript (Materials and Methods section, Lines 371–373 and 386–387).
[Comment] a large space has been left in both introduction and discussion relatively to normalization strategies; this part is rather known and should deserve less attention. ON the contrary, other aspects (such as metabolomics for food traceability, or the relevance of traceability for adzuki beans) would need more attention. Introduction should be re-written.
[Response] We have added the following sentences in the revised manuscript to address this concern: “Metabolomics has been performed to differentiate the geographic origins of many foods, such as green tea, grape berry, wine, Angelica gigas, tobacco, cabbage, olive oil, wheat, pork, tomato, and coffee [15-24]. Several techniques including GC-MS, LC-MS, capillary electrophoresis (CE)-MS, and nuclear magnetic resonance (NMR) spectroscopy allow for accurate determination of the geographical origins of the same variety by analysing differences in chemical composition. Among these techniques, GC-MS represents a relatively robust and inexpensive analytical system. Previously, we used GC–time-of-flight (TOF) MS to investigate the possibility for metabolic discrimination between Tagetes cultivars [25]. Putri et al. [21] reported an application of non-targeted GC-MS metabolomics for the discrimination analysis of geographical origin of coffee samples.” (Lines 60–69)
[Comment] why GC/TOF was chosen for this purpose? I expect that LC/QTOF targeting secondary metabolism would have been more effective in discrimination, because secondary metabolites are more affected by environment
[Response] As mentioned by the reviewer, plant secondary metabolism is influenced by various environmental factors, and LC-MS allows simultaneous detection of a diverse group of secondary metabolites. However, GC-MS represents a relatively robust and inexpensive analytical system. Recently, Putri et al. (2019) reported an application of non-targeted GC-MS metabolomics for the discrimination analysis of geographical origin of coffee samples (Metabolomics, 15, 126). In addition, GC-MS commonly provides profiles of primary metabolites with good reproducibility. We have determined primary metabolism interplay using GC-TOFMS–based metabolite profiling in oval- and rectangular-shaped Chinese cabbage (Foods, 2019 8, 587). Furthermore, Zhao et al. (2015) successfully used GC-MS metabolomics for the investigation of geographical location–associated primary metabolic changes in tobacco plants (Scientific Reports, 5, 16346). Therefore, in this study, we used GC-TOFMS–based metabolite profiling combined with chemometrics to investigate biochemical reactions as well as to construct a discrimination model of geographical origin of adzuki beans samples, for the first time. We have included the above statements in the Introduction section of the revised manuscript (Lines 82–83 and 86–90).
[Comment] the choice of samples is questionable: korean samples are from the same site, i.e. they are not affected by pedo-climatic conditions, whereaas chinese samples are. On the contrary, cultivar is unknown for chinese samples but it is known for korean specimens.
[Response] More than of 80 % of total adzuki bean consumed in Korea has been imported from China. Identification of the geographical origin of adzuki bean is critical because some local vendors deceive customers regarding cultivation origin for economic reasons. Thus, Chinese adzuki bean samples (C1–C6) harvested from two regions (Harbin, Heilongjiang province and Yanji, Jilin province) in November 2017 were purchased from local markets in Xinzhou and Jiangxia (Wuhan city) provinces, China. (Lines 337–339)
However, these sampling locations are poor. An interesting aspect for future research is to clarify the genotype × environmental interactions on the phytochemical composition in adzuki bean. Thus, future work involving larger sample size from various producing region will be very important for geographical discrimination of adzuki bean. We have included the above statements in the Conclusions section of the revised manuscript (Lines 326–330).
[Comment] only 19 out of 36 compounds were confirmed by standard.. this sounds a little limiting, do the authors have an explanation? the ID process should be reported (monoisotopic mass? monoisotopic + isotope ratio?...)
[Response] We confirmed 36 compounds by standards (MSI level 1) in targeted metabolite profiling. ChromaTOF software (LECO, St. Joseph, USA) was used to identify the hydrophilic compounds in adzuki beans. The Chroma TOF software package was used to extract raw peaks, filter and calibrate data baselines, align peaks, perform deconvolution analysis, identify peaks, and integrate peak areas. (Lines 368–370) Similarity index of 70 % or more was chosen as the cut-off value because it afforded 100 % accuracy in analyte identification based on our previous experience (confirmed by co-injection of commercial standards) (Metabolomics, 2019, 15, 21). (Lines 373–375)
[Comment] a biochemical discussion on the discriminant metabolites could be added hierarchical clustering might help dissecting cultivar effects from origin effects, once cultivar type is known why OPLS-DA was not the choice, in place of PLS-DA? in SIMCA you have this possibility.. this would help in limiting non predictive variability and offers the possibility of CV-ANOVA validation as well as permutation testing for overfitting
[Response] We thank the reviewer for this comment. As suggested, we performed OPLS-DA, CV-ANOVA, and permutation test (Lines 274–289, Figure S5). We have added a biochemical discussion on the discriminant metabolites in the revised manuscript as follows: “A clear distinction in malic acid and citric acid concentration between Korea and China samples was observed. Several investigations have reported that malic acid and citric acid in plants varied in different cultivation regions [17-20,41]. The differences in environmental conditions might be the main factors in the different metabolite levels of different geographical origins. Pereira et al. [42] reported that malate content in grape berry was significantly related to light exposure. Malic enzyme activity increases between 10 and 46 °C. A study concerning the metabolic responses of tobacco plants to the environment has demonstrated that the intermediates of the TCA cycle display clear negative correlation with environmental factors such as rainfall and temperature [20]. Therefore, this study suggests that malic acid and citric acid could be key metabolites for regional discrimination.” (Lines 292–301) Additionally, in order to discuss the biochemical relationship on the discriminant metabolites, we performed Pearson’s correlation analysis and hierarchical clustering analysis with 36 metabolites identified by targeted metabolites profiling from adzuki beans. Figure 6 shows the two clusters, one that clustered organic acids, sugars and shikimic acid pathway–related metabolites and another that grouped amino acids and sugar alcohols. The organic acids (such as citric acid, malic acid, oxalic acid, and succinic acid) and sugars (sucrose, glucose, fructose, and galactose) were important metabolites related to energy metabolism (glycolysis and the TCA cycle). Organic acids related to the TCA cycle are produced during photosynthesis, and they serve as carbon skeletons for amino acid biosynthesis and light-harvesting, respectively [18]. Thus, negative relationships were observed between carbon-rich primary metabolites and nitrogen-containing metabolites. (Lines 301–310)
[Comment] line 217: silanol is likely an artefact of derivatization
[Response] We removed several artefacts of derivatization, including silanamine; 2-(3-(2-tetrahydropyranyloxy-1-propynyl)benzeonitrile; methyltris(trimethylsiloxy)silane; acetyl chloride; 4-amino-3-mercapto-1,2,4-triazole; cyclotetrasiloxane;. 2-(4-benzoylphenoxy)ethyl n-ethylcarbamate; fluorophosgene; silanol; disiloxane; and (cis)-3-phenyl-1-vinylcyclobutanol.
[Comment] several errors in language are present throughout the whole manuscript, please revise for English
[Response] The entire manuscript has been edited for language by a professional editing company, Editage (www.editage.co.kr; Job Code: EKMZY_19_2).
Round 2
Reviewer 1 Report
This manuscript has been revised and improved based on reviewers’ comments. Relevant piece of information has been added, especially for the discussions about predictive model validation and the roles of citric acid and malic acid. Appropriate references (citing papers) are also added for the discussion.
Reviewer 2 Report
although sample population is still rather limited and genotype x environment interaction not adressed, the manuscript amended previous flaws